# Prospective, single UK centre, comparative study of the predictive values of contrast-enhanced ultrasound compared to time-resolved CT angiography in the detection and characterisation of endoleaks in high-risk patients undergoing endovascular aneurysm repair surveillance: a protocol

Iain Nicholas Roy,[1,2] Tze Yuan Chan,[3] Gabriela Czanner,[1,4] Steve Wallace,[2] Srinivasa Rao Vallabhaneni[1,2]

For numbered affiliations see end of article.

**Correspondence to**
Mr Iain Nicholas Roy;
iain.roy@liverpool.ac.uk

## ABSTRACT

**Introduction** Diagnosis of endoleaks is imperative to prevent failure of endovascular aneurysm repairs (EVARs). The gold standard for diagnosis of endoleaks is catheter-directed subtraction angiography, which is not a practicable choice for surveillance. CT angiography (CTA) is the historical surveillance modality of choice. Concerns over cost, potential nephrotoxicity of contrast agents and repeated radiation exposure led to colour duplex ultrasound scan (CDUS) becoming an established alternative. CDUS has a lower sensitivity and specificity for endoleaks detection compared to CTA. Contrast-enhanced ultrasound scan (CEUS) represents an improvement of ultrasound imaging but comparisons against CTA report widely varying results, likely due to technical factors of CEUS and limitations of single-phase CTA. The development of time-resolved CTA (tCTA) offers timing information that much more closely mirrors the dynamic information available from CEUS. Theoretically, these two imaging modalities have the best potential for diagnostic accuracy. The aim of this study will be to compare CEUS to tCTA and investigate the utility of other measurements available from tCTA.

**Methods and analysis** This is a prospective, single UK centre, comparative study of paired binary diagnostic imaging modalities. Patients identified in routine post-EVAR surveillance as at risk of having a graft-related endoleak will undergo a CEUS and tCTA on the same day. This will allow the first comparison of CEUS to a semidynamic form of CTA. CEUS sensitivity and specificity to endoleak detection will be calculated.

**Ethics and dissemination** The study has achieved ethical approval. We hope the results will define the diagnostic accuracy of CEUS in comparison to a semidynamic form of CTA, representing a methodological improvement from previous studies. Results will be submitted for presentation

### Strengths and limitations of this study

► First comparison of contrast-enhanced ultrasound to a dynamic form of CT imaging, representing a methodological improvement.
► Both imaging modalities occur on the same day removing changing findings as a confounding factor.
► Primary outcome set as type I/III endoleak (most clinically significant) rather than all endoleaks.
► Appropriately powered but small study.
► Single-centre study.

at national and international vascular surgeryandradiology meetings. The full results are planned to be published in a medical journal.

**Trial registration number** NCT02688751.

## INTRODUCTION
### Background and rationale

Endovascular aneurysm repair (EVAR) is the intervention of choice to treat abdominal aortic aneurysms (AAA).[1] In comparison to open surgical repair, EVAR confers a reduction of mortality lasting into the short to intermediate term.[2] However, EVAR is associated with complications which sometimes require secondary interventions in order to maintain efficacy of EVAR. This has been recognised since the inception of the technique and confirmed in observational studies as well as randomised controlled trials.[3–6] Therefore, periodic surveillance imaging is recommended for life following EVAR.[7 8] The

importance of post-EVAR surveillance remains enduring, its value further highlighted by a recently published analysis of 15-year follow-up after EVAR.[2]

The most common complication in EVAR surveillance is an endoleak,[9] which is 'persistent blood flow within the aneurysm sac but outside the stent graft'.[10] Endoleaks are classified based on the source of blood flow,[10] but can be grouped into stent graft related (types I and III) and type II (non-stent graft related) endoleaks. Stent graft-related endoleaks generally transmit high pressure causing a high risk of aneurysm expansion/rupture (treatment failure).[11] [12] In contrast, type II endoleaks generally run a benign course, particularly in the absence of aneurysm expansion.[12] With regard to endoleak imaging, high sensitivity of detection and high specificity of characterisation improve diagnostic utility of surveillance, in particular with an emphasis on distinguishing stent graft-related endoleaks from type II endoleaks. Digital subtraction angiography in multiple planes and a high frame rate of acquisition is the gold standard of endoleak imaging; high frame rates to demonstrate endoleak heamodynamics for better characterisation. However, this modality is not tenable to be used as surveillance imaging.

Historically, EVAR surveillance was undertaken using CT angiography (CTA). Arterial phase CT was the most frequently used modality although, selectively or routinely additional phases are used such as unenhanced, venous phase and even delayed phase. Concerns over cost, use of potentially nephrotoxic contrast agent and repeated radiation exposure led to alternative imaging modalities being investigated and implemented in surveillance regimens. Colour duplex ultrasound scan (CDUS) is the most widely used imaging modality currently.[13] CDUS is reported to have a lower sensitivity and specificity to detect stent graft-related endoleaks compared with CTA.[14]

Contrast-enhanced ultrasound scan (CEUS) has been investigated as an adjunct to CDUS in the hope of improving sensitivity to endoleak detection. CEUS involves intravenous injection of a microbubble contrast which remains in the blood, allowing improved detection of endoleaks, particularly with contrast coherent ultrasound imaging. CEUS also allows continuous (dynamic) or real-time monitoring of the aneurysm and endoleak as the contrast agent arrives into the endoleak. Modern microbubble agents are expired by the respiratory system, thus avoiding nephrotoxicity. A recent review of 30 222 administrations of a CEUS contrast agent demonstrated a low adverse reaction rate of 0.020%.[15] CEUS also obviates the radiation exposure associated with CTA. 3D acquisition and reconstructions of CEUS scans are possible[16]; however, this development in the technology is currently limited to single phases of acquisition and therefore losses the dynamic information available in standard CEUS.

Systematic review of the diagnostic accuracy of CEUS (in comparison to CTA) in detection of any endoleak revealed variable sensitivity ranging from 67% to 100% and specificity ranging from 79% to 100%.[17] The value of this review is compromised by heterogeneity of contrast material used and non-reporting of ultrasound imaging technique used, specifically whether colour duplex combined with contrast or harmonic CEUS imaging was used. There are potentially additional reasons for variability in the reporting such as operator dependence, quality of equipment used as well as body habitus of the patient. The consideration that neither CEUS nor single-phase CTA represent gold standard of endoleak diagnosis led to the metanalysis[14] [17] adopting a bivariate model of analysis. This approach does not compensate for the lack of gold standard comparator as it favours any modality producing false positives. Comparison against the gold standard has never been established for either modality.

### Time-resolved CTA

Time-resolved CTA (tCTA) was first described for endoleak detection in 2010[18]. The single arterial phase is replaced by multiple phases in tCTA, which are typically of lower radiation dose, thus offering dynamic observations of endoleaks, such as flow direction and filling speed, while still retaining many of the advantages of CTA (3D reconstruction, etc) which closely mirror the advantages of the multiplanar digital subtraction catheter angiography. The multiple phases of tCTA can be achieved by broad CT detectors and a static patient in the Fowler position[19] or rapid shuttling of a supine patient through a standard detector.[20] [21] Sufficient amount of measurements regarding filling patterns of endoleaks on tCTA[21] are now available to be able to replace a standard arterial phase in CTA with a tCTA that is aimed at detecting stent graft-related endoleaks, without increasing radiation exposure for the patient. Now timings and interpretation of tCTA are understood it is timely to do a comparison of CEUS to the improved comparator of tCTA. This overcomes the limitations of previous studies by comparison of CEUS to a (semi)dynamic form of CTA imaging as a gold standard.

### METHODS

### Study design

This is a prospective single-centre comparative study of paired diagnostic imaging modalities, designed to comply with the 'standards for reporting diagnostic accuracy studies'.[22] [23] Participants will be recruited from a city-wide vascular service in the UK. The service is arranged in a hub and spoke configuration locally and regularly accepts tertiary referrals for complications of previous aortic surgery at other centres. EVAR surveillance is predominately undertaken using CDUS and plain radiography.[24]

### Participants

Potential participants are referred to the study by their vascular surgeon when they require a CTA for further investigations of an endoleak following routine EVAR surveillance and meet inclusion criteria (box 1). They are typically patients with a suspicion of a graft-related endoleak or aneurysm expansion on CDUS surveillance. They are then assessed by the study for eligibility based on

## Box 1 Inclusion criteria

► Aged 18 or over.
► Able to give informed consent.
► Undergone an endovascular aneurysm repair (EVAR) of infrarenal abdominal aortic aneurysm.
► Planned for CT angiography of EVAR.

exclusion criteria (box 2) and approached to participate in the study, if appropriate. Patients are approached by an investigator and those who give their written informed consent to participate will be enrolled in the study.

### Test methods

Participants attend and have a CEUS (index test) and tCTA (reference standard) on the same day. This represents a change from standard care for participants who would otherwise have a CEUS and triple phase CTA (non-contrast, arterial (20s) and delayed venous phase (90s)) often on separate days. Participants will also be asked a short number of closed questions to assess functional status and cardiac function (online Supplementary material). On completion of the CEUS and tCTA, participants are returned to their referring surgeons, with clinical reports of the studies for ongoing care. The study team follows the participants' further treatment and investigation until the end of the study.

### Contrast-enhanced ultrasound

CEUS is performed in combination with a standard CDUS in our institution. It is reported as a binary test yielding two values: present or absent for each endoleak type. It is conducted by an experienced clinical vascular scientist with extensive involvement in scans for EVAR surveillance. It is performed on a Philips IU22 ultrasound machine (Philips, Amsterdam, Netherlands), using the a 2–5 MHz abdominal curved array probe. Grey-scale images of the aneurysm neck (when possible), iliac seal zones and maximum aneurysm dimensions are obtained

## Box 2 Exclusion criteria

► Unable to receive CT angiography (CTA) contrast
  – Allergy.
  – Insufficient renal function for standard outpatient contrast study (estimated glomerular filtration rate <45).
  – Overactive thyroid gland.
► Unable to receive contrast-enhanced ultrasound scan contrast
  – Previous reaction to Sonovue® (ultrasound contrast).
  – Allergy to sulphur hexafluoride (used in electrical industry in circuit breakers, switch gears and electrical equipment).
  – Recent acute coronary syndrome or unstable angina, typical angina at rest or frequent or repeated angina/chest pain—all within previous 7 days.
  – Recent coronary intervention.
► Previous embolisation of artery in region of endovascular aneurysm repair (affects imaging quality).
► Body mass index >30 (affects imaging quality).

and measured in maximum anteroposterior and mediolateral dimensions. Note is made of the echogenicity of thrombus within the aneurysm sac. Using colour flow imaging and spectral Doppler, waveform characteristics and velocities are recorded in the common femoral arteries. The stent graft is interrogated using colour and spectral Doppler to ascertain patency and flow heamodynamics of the neck, main body and both limbs. Any abnormalities in these parameters are reported. Colour Doppler is used to detect any endoleak. If present, its type, point of inflow, point of outflow and flow dynamics (using sectoral Doppler) are reported.

Optimum views of the area of concern are obtained, prior to contrast injection, using appropriate machine set-ups and controls as determined by the operator. Sulphur hexafluoride microbubble contrast (2.4 mL; SonoVue, Bracco, Milan, Italy) is injected followed by 10 mL of sodium chloride 0.9%; the on-screen timer is started at the start of the injection. Flow direction and filling time ideally should be determined and anatomy of the endoleak established by interrogation. Passive elimination of the contrast agent is allowed to occur and the process repeated for a second injection.

CEUS scan will be reported by the performing vascular scientist to the data point recorded in the data collection pro forma (online Supplementary material), in addition to any clinically relevant points. The vascular scientist will be blinded to the concurrent tCTA at the time of reporting, although will be aware of the previous findings on EVAR surveillance.

### Time-resolved CTA

tCTA is performed on a Siemens Definition AS+scanner (Seimens, Munich, Germany), in our institution. Participants are positioned supine with arms raised above their head. The contrast injector is connected to a 20 G (or larger) intravenous catheter in an anterior cubital fossa vein. A standard topogram scan is performed. Unless not required, a non-contrast scan is performed. The maximum length that can be covered for the time resolution required is 27 cm. This is centred over the EVAR stent graft. Abdominal guides are placed at upper aspect of diaphragm and common femoral arteries for venous phase of scan. Abdominal aorta, just proximal to EVAR graft is selected as trigger area for time-resolved phases.

Contrast is injected, using an auto injector, at 4 mL/s for 96 mL. Participants are asked to adopt shallow breaths and not hold their breath. The time-resolved phase is triggered by a Hounsfield unit (HU)>90 in trigger area. Phases occur at 2.5, 5, 7.5, 10, 15, 20 and 25 s following the automatic trigger; these occur in craniocaudual acquisition, except the 5 and 7.5 s phases which occur in a caudocranial direction. The venous phase is taken in full inspiration and is acquired 75 s following the trigger. Tube setting and calculated predicted radiation exposure are presented in table 1.

All tCTA scans will be reported by a single consultant vascular radiologist. This reporter will be blinded to the

**Table 1** Settings and radiation exposure from arterial and time-resolved phases of CT angiography

|  | Arterial phase (outside study) | Time-resolved phases (inside study) |
|---|---|---|
| Tube voltage | 120 kV | 80 kV |
| Tube current | 230 mA (effective current–scanner automatically varies) | 120 mA |
| Scan length | Variable (dependant on body length) | 27 cm |
| Number of Phases | One | Seven |
| Expected DLP | 599.6* | 78.9 mGy/cm per phase (552.3 mGy/cm for time-resolved phase) |

*Average DLP used for an arterial phase scan in all CT angiography scans in Royal Liverpool Hospital in month of July 2015.
DLP, Dose Length Product.

results of the CEUS and collect data to the pro forma. It will be reported as a binary test yielding two values: present or absent for each type of endoleak.

## Outcome measures

The primary outcome is:

1. The predictive values of CEUS in comparison to tCTA (as comparator) to detect stent graft-related endoleaks.

   Secondary outcomes are:

1. Any adverse events during CEUS or tCTA.
2. Predictive values of CEUS in comparison to tCTA to detect type II endoleaks.
3. Predictive values of both tCTA and CEUS in predicting final endoleak diagnosis (following any further investigations).
4. Predictive values of both tCTA and CEUS in predicting need for a secondary intervention.
5. Evaluate the association between CEUS temporal delay (difference between contrast in endograft and contrast in endoleak) and evaluate its ability to improve the differentiation of endoleak type.
6. Evaluate the association between 'CEUS contrast in endoleak' to 'tCTA contrast in endoleak' and assess potential as predictive tool, for optimum timing of CTA phases.

   Analysis plan

Associations will be established/refuted with summary statistics and graphical analysis and appropriate further statistical testing within the framework of logistic regression. If association can be established, then predictive modelling will be undertaken. The agreement between CEUS and tCTA will be evaluated with Kappa statistic.[25] Sensitivity/specificity will be calculated along with binomial exact 95% CIs and leave-one-out cross-validation. We

will also report the positive predictive value and negative predictive value.

The power calculation[26] showed the required sample size to be 74. This was calculated based on a prevalence of stent graft-related endoleaks of 11% as demonstrated on previous tCTA studies of endoleaks. It was powered to detect a predicted sensitivity of 0.95 with a tolerated CI of ±0.15. The study commenced recruitment in February 2016 and is ongoing.

Monitoring and data

All patients referred to the study are recorded in the screening log. The sponsoring trust will provide governance oversight and annual reports to the ethics committee will provide ethical overview. Radiation exposure of participants will be monitored and reported to the sponsors and approving research ethics committee. No interim analysis is planned.

On completion of the study, all identifiable patient demographics will be removed and anonymised data will be stored to allow for future analysis of unforeseen benefits. During the study, data are stored in a secure manner within the host institutions data management processes. Access is restricted to the investigators and for audit by the sponsor.

Ethics and dissemination

The main ethical consideration is the change of care from CTA to a tCTA; this was felt to be appropriate in the context of informed consent, now it can be performed without increasing radiation exposure. Ethical approval (15/NW/0908) was granted by a National Health Service Research Ethics Committee. On completion of analysis, we expect to publish in a medical journal, along with the anonymised data set and present the findings widely.

**Author affiliations**
[1]Institute of Ageing and Chronic Disease, University of Liverpool, Liverpool, UK
[2]Liverpool Vascular and Endovascular Service, Royal Liverpool University Hospital, Liverpool, UK
[3]Department of Radiology, Royal Liverpool University Hospital, Liverpool, UK
[4]Department of Biostatistics, University of Liverpool, Liverpool, UK

**Contributors** IR developed and wrote the protocol. TC reviewed and revised the radiological/imaging elements of the protocol. SW reviewed and revised the ultrasound elements of the protocol. GC reviewed and revised the statistical analysis elements of the protocol. SV conceived the project and supervised IR in developing the protocol.

**Funding** This study is sponsored by Royal Liverpool and Broadgreen University Hospitals Hospitals NHS Trust (Ref: 5083) and supported by use of staff and resources from The Royal Liverpool University Hospital, Liverpool, UK, and The Institute of Ageing and Chronic Disease, The University of Liverpool, UK.

**Competing interests** None declared.

**Patient consent** Detail has been removed from this case description/these case descriptions to ensure anonymity. The editors and reviewers have seen the detailed information available and are satisfied that the information backs up the case the authors are making.

**Ethics approval** NHS Research Ethics Committee.

**Provenance and peer review** Not commissioned; externally peer reviewed.

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
