## [Reviewer comments · BMJ Open]

ARTICLE DETAILS

TITLE (PROVISIONAL)	Protocol: A Prospective, Single UK Centre, Comparative study of the Predictive Values of Contrast Enhanced Ultrasound compared to Time-Resolved Computer Tomography Angiography in the Detection and Characterisation of Endoleaks in High Risk Endovascular Aneurysm Repair Surveillance Patients
AUTHORS	Roy, Iain; Chan, Tze; Czanner, Gabriela; Wallace, Stven; Vallabhaneni, Srinivasa

VERSION 1 – REVIEW

REVIEWER	Raymond Ashleigh University hospital of South Manchester UK
REVIEW RETURNED	18-Dec-2017

GENERAL COMMENTS	Important clinical question that may change practice in UK vascular centres
---

REVIEWER	Gareth Harrison South Mersey Arterial Centre, UK
REVIEW RETURNED	29-Dec-2017

GENERAL COMMENTS	please define which patients are high risk for endograft issues to be included in the study. Is the radiation dose with tCTA greater than conventional CTA?
--

REVIEWER	Professor Charles McCollum Professor of Surgery Cardiovascular Institute University of Manchester. Honorary consultant surgeon, Manchester foundation trust
REVIEW RETURNED	08-Jan-2018

GENERAL COMMENTS	The authors propose a study comparing contrast enhanced ultrasound (CEUS) with time-resolved CT angiography (t CTA). References they quote do not confirm that t CTA is more accurate than single arterial phase CTA which is widely used by most centres at the moment. As t CTA is not widely used, why compare this to CEUS as the gold standard; it needs to be established as the gold standard before this comparison is relevant.
--

	The authors have ignored two recently published papers, although from the same group the second being a larger study than the first, showing that contrast enhanced 3-D tomographic Ultrasound Appears to be the optimal and least invasive way to detect endoleak following EVAR. Is it really relevant to be exploring tCTA which is expensive, involves ionising radiation and cumulatively nephrotoxic x-ray contrast media under these circumstances? The later paper did include comparisons with standard CEUS in large numbers than in the proposed study: Abbas et al. Eur J Vasc & Endovasc Surg 2014;47: 487-92. Lowe et al. J Vasc Surg 2016; 65: 453-59.
--	--

VERSION 1 – AUTHOR RESPONSE

Editorial Requirements:

Please include the study location define EVAR and in the title

We have now included that it is a UK centre and defined EVAR in the title.

EVAR is already defined in the Abstract and main article

We have now specified it is a UK single centre study in the abstract and main article and the individual centre is identifiable from the sponsor and authors institution.

Reviewer: 1

Important clinical question that may change practice in UK vascular centres

We thank the reviewer for their positive feedback

Reviewer: 2

please define which patients are high risk for endograft issues to be included in the study.

Is the radiation dose with tCTA greater than conventional CTA?

These are defined in Table 1&2 but we have added a sentence in the first paragraph of page 7 to summarise the patients being recruited.

The fact that tCTA can now be performed without an increase in radiation exposure is clearly stated in the text and demonstrated by the figures for the Dose length Product (DLP) in table 3. If the editors feel this isn't conveyed well we are happy to take direction on changes, but on re-reading the manuscript we believe it is plainly stated.

Reviewer: 3

[1] The authors propose a study comparing contrast enhanced ultrasound (CEUS) with time-resolved CT angiography (tCTA). References they quote do not confirm that tCTA is more accurate than single arterial phase CTA which is widely used by most centres at the moment. As tCTA is not widely used, why compare this to CEUS as the gold standard; it needs to be established as the gold standard before this comparison is relevant.

Our study uses tCTA as the reference standard and CEUS as the comparator. We agree that there is no direct comparative evidence definitively proving tCTA is more accurate than single arterial phase CTA, this is because such a study would likely require the same patients to undergo tCTA and CTA which would likely be deemed unethical in light of the below knowledge.

tCTA has demonstrated the perfusion patterns of endoleaks, which in turn demonstrates that a single phase CTA performed at any particular time point will not demonstrate all endoleaks. A single phase CTA, by definition, can also not define flow direction thereby limiting its ability to differentiate different types of endoleaks once they are detected. We discuss this in the manuscript in the simplified terms of dynamic versus static [non-dynamic] imaging.

The use of single arterial phase CTA as a "gold" standard, when the comparator is a dynamic form of imaging such as CEUS is questionable in the light of the above knowledge and has led to the potential underestimation of CEUSs predictive values – the objective of this study is to demonstrate

these in relation to graft related endoleaks (the most clinically significant) with a dynamic form of imaging as the reference standard. We are not advocating tCTA as a replacement for CTA or CEUS but are using it as the reference standard, to define the true predictive values of CEUS.

[2] The authors have ignored two recently published papers, although from the same group the second being a larger study than the first, showing that contrast enhanced 3-D tomographic Ultrasound Appears to be the optimal and least invasive way to detect endo-leak following EVAR. Is it really relevant to be exploring tCTA which is expensive, involves ionising radiation and cumulatively nephrotoxic x-ray contrast media under these circumstances? The later paper did include comparisons with standard CEUS in large numbers than in the proposed study.

We are exploring the predictive values of CEUS not tCTA. 3D CEUS is conceded to be a novel technique in your referenced papers, all be it, one which holds considerable promise. Our concern is that it takes the dynamic information out of CEUS as the 3D reconstructions require a pass over the patient to acquire images and time to reconstruct the 3D image. The paper does include a comparison of CEUS with single arterial phase CTA which is subject to the same limitations as above.

Regarding 3D CEUS, currently we can have the dynamic filling information of 2D CEUS followed by a 3D render of the anatomy but can't have both at the same time which is what tCTA offers. We hope to define if timing information on CEUS could be used to better time a single phase CTA for institutions that don't have 3D CEUS available, as is listed in our secondary objectives. We have added a sentence regarding 3D CEUS to the 2nd paragraph on page 6 and look forward to the evolution of this exciting technology.

VERSION 2 – REVIEW

REVIEWER	Gareth Harrison Countess of Chester Hospital, UK
REVIEW RETURNED	06-Feb-2018
GENERAL COMMENTS	Happy from my previous comments